# Commutation Torque Ripple Reduction Strategy of Brushless DC Motor Drives Based on Boosting Voltage of DC-Link Small Capacitor

**DOI:** 10.3390/mi13020226

**Published:** 2022-01-29

**Authors:** Xinmin Li, He Yuan, Wei Chen, Lihong Yu, Xin Gu

**Affiliations:** 1School of Electrical Engineering, Tiangong University, Tianjin 300387, China; lixinmin@tju.edu.cn (X.L.); yuanhe@tiangong.edu.cn (H.Y.); chen_wei@tiangong.edu.cn (W.C.); guxin@tiangong.edu.cn (X.G.); 2College of Electronic Information and Automation, Civil Aviation University of China, Tianjin 300300, China

**Keywords:** brushless DC motor (BLDCM), DC-link, small capacitor, boost control, commutation torque ripple reduction

## Abstract

Based on the brushless DC motor system with DC-link small capacitance powered by a single-phase AC power source, a boosting DC-link voltage strategy to reduce the commutation torque ripple of brushless DC motors is proposed in this paper. The control strategy utilizes the special topology of the motor system to boost the DC-link capacitor voltage in a specific zone during the non-commutation period. During the commutation period, the high voltage of the DC-link capacitor is released to meet the voltage requirement of the brushless DC motor during commutation. In order to reduce the commutation torque ripple and ensure the normal operation of the brushless DC motor, each rectifier cycle is divided into three zones according to the characteristics of the periodic change of the rectifier output voltage. Different operation modes are proposed for different zones. In DC-link capacitor boost voltage mode, the DC-link capacitor boosts the voltage to meet the voltage of the motor demand during the commutation period for achieving the purpose of reducing the commutation torque ripple. In this paper, the controller of the brushless DC motor system is designed and the experimental platform is built. The experimental results verified the correctness of the theoretical analysis and the feasibility of the proposed method.

## 1. Introduction

Because of its advantages of simple structure, large output torque, and high power density compared with traditional brush DC motors, brushless DC motors (BLDCM) have been widely used in aerospace, industrial transmission, marine exploration and other fields [1,2,3]. However, the BLDCM usually adopts a two-phase conducting mode, which will result in torque ripple during the commutation period, and the torque ripple can reach more than 50% of the average load torque. The large noise and vibration produced by commutation torque ripple will affect the normal operation of the load equipment, and seriously restrict the application of the BLDCM under high-precision and high-stability operating conditions [4,5,6].

In order to reduce the impact of commutation torque ripple on the operation of BLDCM systems, a series of studies has been carried out to reduce the commutation torque ripple of BLDCMs by relevant scholars. The main reduction methods include the PWM method [7,8], the current predictive control method [9,10], the direct torque control method [11], etc. However, the above-mentioned suppression methods have problems with practical applications, such as switch frequently between high and low speed, saturation of the output signal of the PWM modulator, increased commutation time, difficulties in obtaining accurate motor models, and poor dynamic performance.

Therefore, some scholars have proposed strategies of using DC voltage adjusting technology to reduce the commutation torque ripple [12,13,14]. The voltage required during the commutation period of the BLDCM can be achieved by boosting the DC bus voltage, and the commutation torque ripple can be reduced without increasing the commutation time. In [12], a control strategy that uses a SEPIC converter which outputs a high voltage during the commutation period to reduce the commutation torque ripple was proposed. A Z-source converter was used as the front-end topology to boost the voltage by using a shoot-through vector in [13]. In [14], a strategy based on the boost topology without an inductor could achieve the effect of commutation torque ripple reduction without additional boost circuits.

The above commutation torque ripple reduction strategies are all suitable for operating conditions powered by DC power source. However, in some applications, only an AC power source can supply energy to the BLDCM. In a BLDCM system powered by a single-phase AC power source, it is often necessary to convert AC power into DC power through a rectifier bridge, and the DC-link side needs to set a large electrolytic capacitor to ensure the stability of the DC power source. However, electrolytic capacitors have problems with poor stability and short service life, so the structure of a thin film capacitor is favored [14,15]. However, a thin film small capacitor cannot effectively stabilize DC-link voltage, which will cause motor current ripple or even cut off, and could even seriously affect the normal operation of the motor. For this reason, studies [16,17] proposed a new DC-link topology with a switch tube and a small capacitor in series. By using this topology structure, the DC-link voltage can be adjusted to maintain the stability of the motor current, and then the smooth operation of the motor is realized. Based on the topology structure proposed in [16,17,18] proposed a special spider algorithm to maintain the stability of motor current and reduce the motor torque ripple.

As the main component of motor torque ripple, commutation torque ripple affects the stable operation of the motor to a certain extent and restricts the development of the motor system. Therefore, the study of commutation torque ripple has practical application value. However, for BLDCM systems with DC-link small capacitance powered by single-phase AC power sources, no scholars have studied the motor commutation torque ripple. At present, related studies only focus on the torque fluctuation caused by DC-link voltage instability. In this paper, based on a BLDCM system with DC-link small capacitance powered by single-phase AC power source, a boosting DC-link voltage control strategy to reduce commutation torque ripple is proposed for the first time.

## 2. The Brushless DC Motor (BLDCM) Drive System with Small DC-Link Capacitor 

### 2.1. DC-Link Topology

The equivalent circuit of brushless DC motor (BLDCM) system of with a small DC-link capacitor is shown in Figure 1. The system is made up of an AC power source, a single-phase diode rectifier bridge, inverter, a brushless DC motor and a DC-link circuit, which is further composed of a small film capacitor *C* and a switch T, in series. T and T_1_–T_6_ use IGBT, and D and D_1_–D_6_ are continuous diodes connected in anti-parallel on the IGBT. The DC-link capacitor voltage is set as *u*_cap_, the AC voltage instantaneous value is set as *u*_s_, the DC-link voltage is set as *u*_d_link_, the motor phase inductance is set as *L*, and the motor phase resistance is set as *R*. In Figure 1, N is the motor neutral point, O is the voltage reference point, and the arrow direction is the positive direction of current.

If the instantaneous value of the DC-link voltage and AC voltage meet *u*_cap_ < |*u*_s_|, the diode D in Figure 1 will be set to on to charge capacitance *C*. As the capacitance of the DC-link capacitor is so small, the lagging of capacitor voltage can be ignored and the DC-link voltage and AC voltage instantaneous value meet *u*_cap_ ≥ |*u*_s_| at all times. In this case, the on–off pattern of switch T can make the DC-link voltage change. When switch T is on, the motor energy is supplied by the DC-link capacitor, as shown in Figure 2a. When switch T is off, the power of the motor is supplied by the output of the diode bridge rectifier, as shown in Figure 2b. The DC-link voltage in the two states can be expressed as
(1)$$u_{d\_\mathrm{link}}=\left\{ \begin{array}{l} u_{\mathrm{cap}}\;(\mathrm{Switch}\;T\;\mathrm{ON})\\ |u_{s}|\;(\mathrm{Switch}\;T\;\mathrm{OFF}) \end{array}\right.$$

### 2.2. Operation Principles of BLDCM

In the pairwise switching mode, take the “a + b −” commutation period as an example. Assume that phase “a” is the positive conducting phase and that phase “b” is the negative conducting phase. The phase voltage of the two conducting phases can be expressed as:(2)$$\left\{ \begin{array}{l} u_{a}=i_{a}R+L\frac{di_{a}}{dt}+e_{a}+u_{N}\\ u_{b}=i_{b}R+L\frac{di_{b}}{dt}+e_{b}+u_{N} \end{array}\right.$$
where *u*_a_, *u*_b_ are the voltages of phase a and phase b, respectively. As shown in Figure 1, the phase current meets the requirement of *I* = *i*_a_ = −*i*_b_, and the phase back electromotive forces (EMF) meet *E* = *e*_a_= −*e*_b_. By determining the difference of the two equations in (2), the voltage of phase “a” and phase “b” can be obtained. The phase current of the motor is approximately constant in one commutation cycle. If the drop in the phase inductance voltage of the motor is ignored, the average line-to-line voltage of the conducting two-phase can be expressed as
(3)$$U_{\mathrm{ab}}=2E+2IR$$

The average line-to-line voltage of the conducting two-phase given in maintenance Equation (3) is the necessary condition to ensure that the brushless DC motor can operate smoothly. When the motor speed and load are constant, the average line-to-line voltage of the conducting two-phase remains unchanged. When the motor is in rated operation, Equation (3) can be rewritten as (4), with *E*_N_ as the rated phase EMF and *I*_N_ as the rated current.
(4)$$U_{\mathrm{abN}}=2E_{N}+2I_{N}R$$

If the DC-link voltage *u*_d_link_ cannot meet the conditions shown in Equation (4), the motor phase current will drop significantly or intermittently under rated conditions, which will affect the smooth operation of the brushless DC motor.

The output voltage of the diode rectifier bridge |*u*_s_| and the DC-link capacitor voltage *u*_cap_ waveform of existing control method are shown in Figure 3. Let *U*_m_ be the AC voltage amplitude and *f* be the grid frequency. With the drop in diode voltage ignored, the diode rectifier bridge output voltage amplitude of the rectifier bridge is *U*_m_, and the period is *T*_R_ = 1/2*f*. According to the size relation of |*u*_s_| and *U*_abN_, a rectification period *T*_R_ is divided into three zones: Zone A, Zone B and Zone C.

Zone A: |*u*_s_| ≥ *U*_abN_ and |*u*_s_| increase monotonously.Zone B: |*u*_s_| ≥ *U*_abN_ and |*u*_s_| decrease monotonously.Zone C: |*u*_s_| < *U*_abN_;

To ensure that the brushless DC motor can work continuously within the rated operation range, the DC-link voltage in all three zones must meet the following condition:(5)$$u_{d}{}_{\_\mathrm{link}}\geq U_{\mathrm{abN}}$$

As shown in Figure 3, in one rectification period *T*_R_, the output voltage of the rectifier bridge in Zone A and B |*u*_s_| is greater than *U*_abN_. It can be seen from the second equation of Equation (1) that the switching tube T is kept off in Zones A and B, and the DC-link voltage *u*_d_link_ = |*u*_s_|, the constraint condition shown in Equation (5) can be satisfied. In Zone C, the output voltage of the rectifier bridge |*u*_s_| is less than *U*_abN_, which cannot meet the constraint condition of Equation (5). Therefore, the switch T is on to discharge the DC-link capacitor. At this time, the DC-link voltage *u*_d_link_ = *u*_cap_ so that the constraint conditions shown the Equation (5) are met, which guarantees the smooth operation of the brushless DC motor.

### 2.3. Control Method during Normal Conduction Period

In the paper, three vectors: the effective vector, the charging vector, and the zero vector were used. Below, the three kinds of vectors are defined in detail.

Effective vector: in the operation process of a brushless DC motor, the DC-link power source (rectifier bridge output power or DC-link capacitor) provides energy to the motor. The vector which generates electric current from the DC-link power side flow to the motor side is the effective vector. Under the effective vector, the energy of the DC-link power source needs to be consumed.

Charging vector: in the operation process of a brushless DC motor, the motor carries out reverse charging on the DC-link side. The vector which generates the current flowing from the motor side to the DC-link power side is the charging vector. The DC-link power source stores energy under the charging vector.

Zero vector: in the operation process of a brushless DC motor, the DC-link power source does not provide energy to the motor. The vector which generates the current only continues between the motor and the inverter is the zero vector. Under the zero vector, the energy circulates inside the motor.

According to the characteristics of the rectifier bridge output voltage |*u*_s_|, in order to ensure the smooth operation of the brushless DC motor in the normal conduction period, the DC-link small capacitor motor system has three different working modes: normal operation mode, DC-link boost energy storage mode, and DC-link capacitor voltage step-down mode. Take the “a + b −” conduction period as an example to analyze the three working modes:

A normal operation mode:

In the normal operation mode, the effective vector is *V*_s_, the zero vector is *V*_0_, and the DC-link switch T is in the off state. The vector equivalent circuits of this mode are shown in Figure 4a,b, respectively.

Suppose the duty cycle under the effective vector *V*_s_ is *d*_A_, and the duty cycle under the zero vector *V*_0_ is 1 − *d*_A_. In this case, switch T_1_ is chopping with the duty cycle *d*_A_, and = switch T_6_ is always on. At this time, the two-phase average line voltage *U*_ab_ is:(6)$$U_{\mathrm{ab}}=\left|u_{s}\right|\cdot d_{A}+(1-d_{A})\cdot 0=\left|u_{s}\right|\cdot d_{A}$$

B DC-link boost energy storage Mode:

In order to achieve the effect of reducing torque ripple across the whole range, it can be seen from Figure 3 that the DC-link capacitor is discharged in Zone C, and the capacitor voltage is in a decreasing state. However, it is necessary to ensure that the high voltage required for the commutation period is still satisfied until the end of Zone C. For this reason, we chose to boost the energy storage of the DC-link capacitors in Zone B. In the DC-link boost energy storage mode, the effective vector is *V*_s_, the charging vector is *V*_c_, and the DC-link switch T is off. The vector equivalent electricity in this mode is shown in Figure 5a,b, respectively.

This mode is mainly used for boosting the DC-link capacitor voltage. If the duty cycle set under the effective vector *V*_d_ is *d*_B_, then the duty cycle under the charging vector *V*_c_ is 1 − *d*_B_. In this case, the switches T_1_ and T_6_ are chopping simultaneously with the duty cycle *d*_B_. At this time, the conduction two-phase average line voltage *U*_ab_ is:(7)$$U_{\mathrm{ab}}=\left|u_{s}\right|\cdot d_{B}+(1-d_{B})(-u_{\mathrm{cap}})=d_{B}(\left|u_{s}\right|+u_{\mathrm{cap}})-u_{\mathrm{cap}}$$

C DC-link capacitor step-down mode:

In the DC-link capacitor step-down mode, the effective vector is *V*_e_, the zero vector is *V*_0_, and the DC-link switch T is on. The vector equivalent circuits of this mode are shown in Figure 6a,b, respectively.

In this mode, the DC-link switch T is on and the DC-link capacitor provides energy to the brushless DC motor. Suppose the duty cycle under the effective vector *V*_e_ is *d*_C_, and the duty cycle under the zero vector *V*_0_ is 1 − *d*_C_. In this case, switch T_1_ is chopping with the duty cycle *d*_C_, and switch T_6_ is constantly conducting. At this time, the two-phase average line voltage *U*_ab_ is:(8)$$U_{\mathrm{ab}}=u_{\mathrm{cap}}\cdot d_{C}+(1-d_{C})\cdot 0=u_{\mathrm{cap}}\cdot d_{C}$$

Table 1 lists the states of DC-link switch T and inverter switch T_1_, T_6_ during the “a + b −” normal conduction period. In Table 1, during the period of “a + b −” normal conduction, within the three zones, inverter switch T_1_ is chopping with the duty cycle *d*_A_, *d*_B_, *d*_C_, and inverter switch T_6_ is in a conducting state in the three zones. The DC-link switch T is off in Zone A and B, and on in Zone C.

## 3. Commutation Torque Ripple Reduction Strategy

### 3.1. Causes Commutation Torque Ripple

A BLDCM usually operates in two-phase conducting mode. Due to the existence of the inductance in the motor, the current cannot suddenly change during the commutation period, so the voltage equation of the three-phase winding terminal of the BLDCM can be expressed as follows
(9)$$\left\{ \begin{array}{c} u_{a}=Ri_{a}+L\frac{di_{a}}{dt}+e_{a}+u_{N}\\ u_{b}=Ri_{b}+L\frac{di_{b}}{dt}+e_{b}+u_{N}\\ u_{c}=Ri_{c}+L\frac{di_{c}}{dt}+e_{c}+u_{N} \end{array}\right.$$

Take the “a + b − → a + c −” commutation period as an example for analysis. Now the phase back EMF meets *E* = *e*_a_ = −*e*_b_ = −*e*_c_. Due to the three-phase stator windings connected by star-shaped symmetry, the phase currents meet *i*_a_ + *i*_b_ + *i*_c_ = 0. Furthermore, the electromagnetic torque (*T*_e_) equation during the commutation period can be given by
(10)$$T_{e}=\frac{e_{a}i_{a}+e_{b}i_{b}+e_{c}i_{c}}{\omega_{m}}=-\frac{2Ei_{a}}{\omega_{m}}=\frac{2EI}{\omega_{m}}$$
where *E* is the amplitude of the phase back EMF, *I* is the amplitude of the non-commutation phase current, and *ω*_m_ is the mechanical angular velocity of the BLDCM. From Equation (10), it can be seen that, in a commutation period, *T*_e_ is proportional to the non-commutation current *I*.

Based on Equation (10), it can be seen that the commutation torque ripple can be reduced by keeping the non-commutation phase current stable during the commutation period. Substituting Equation (9) into (10), the average rate of change of the non-commutated phase current *i*_a_ is
(11)$${\frac{di_{a}}{dt}|}_{\mathrm{avg}}=\frac{2u_{a}-u_{b}-u_{c}-(4E+3IR)}{3L}$$

During the actual operation of the motor, in order to make the non-commutation current stable, let the average change rate of *i*_a_ in Equation (11) be 0, and we can obtain
(12)$$2u_{a}-u_{b}-u_{c}=4E+3IR$$

During the “a + b − → a + c −” commutation period, the equivalent circuit diagram is shown in Figure 7. Phase “a” is a non-commutation phase, phase “b” is a non-conducting phase, and phase “c” is a conducting phase. The negative conducting current switches from phase “b” to phase “c”. The equivalent circuit diagram is shown in Figure 7a when switches T_1_ and T_2_ are on. At the same time, *u*_a_ = *u*_dc_link_, *u*_b_ = *u*_dc_link_, *u*_c_ = 0. The equivalent circuit diagram is shown in Figure 7b when the switch T_1_ is off and the switch T_2_ is on. At the same time, *u*_a_ = 0, *u*_b_ = *u*_dc_link_, *u*_c_ = 0.

Assuming that the duty cycle of switch T_1_ is *d*_com_, and the average value of the three-phase winding terminal voltage is
(13)$$\left\{ \begin{array}{c} u_{a}=u_{\mathrm{dc}\_\mathrm{link}}\cdot d_{\mathrm{com}}\\ u_{b}=u_{\mathrm{dc}\_\mathrm{link}}\\ u_{c}=0 \end{array}\right.$$

Substituting Equation (13) into (12), the duty cycle *d*_com_ to maintain a stable non-commutation phase current during the commutation period is
(14)$$d_{\mathrm{com}}=0.5+\frac{4E+3RI}{2u_{\mathrm{dc}\_\mathrm{link}}}$$

In the topology in this paper, the high voltage of the DC-link capacitor is used to maintain the stability of the non-commutation current. During the commutation period, *u*_dc_link_ = *u*_cap_. Equation (14) can be written as
(15)$$d_{\mathrm{com}}=0.5+\frac{4E+3RI}{2u_{\mathrm{cap}}}=0.5+\frac{\pi k_{e}n}{15u_{\mathrm{cap}}}+\frac{3RI}{2u_{\mathrm{cap}}}$$

Since the duty cycle *d*_com_ ∝ [0,1], it can be seen from Equation (15) that the voltage of the DC-link capacitor during the commutation period needs to meet
(16)$$u_{\mathrm{cap}}\geq 4E+3RI$$

### 3.2. Commutation Torque Ripple Reduction Strategy

Based on the BLDCM system with small DC-link capacitance, the voltage of the DC-link capacitor is increased by boosting the voltage during the normal conduction period in this paper, so that the voltage of the DC-link capacitor should always meet the condition of Equation (16). During the commutation period, the high voltage of the DC-link capacitor is used to reduce the commutation torque ripple. We take “a + b − → a + c −” commutation period as an example for analyzing the commutation torque ripple reduction strategy during the commutation period.

After the “a + b − → a + c −” commutation period starts, T_6_ is off, T_2_ is on, and T_1_ is chopping. In order to maintain the stability of the non-commutation current during the commutation period, it can be seen from Equation (16) that T_1_ needs to be chopped with duty cycle *d*_com_, and that both DC-link switch T and inverter side switch T_2_ are on.

Taking the above commutation process as an example, it can be extended to six commutation processes in an electric cycle. According to the motor position information collected by the Hall sensor, two switches on the inverter side during the commutation period are on. One of switch is chopping with duty cycle *d*_com_, and the other is always on. Meanwhile, switch T of the DC-link is always on, and the current ripple is reduced by the higher voltage of the DC-link capacitor during the commutation period.

### 3.3. Controller Design

In order to reduce the commutation torque ripple of a BLDCM system with a DC-link small capacitor and ensure the normal operation of the motor, the controller of the proposed control strategy is designed. Figure 8 is the structure of the control system in the proposed strategy. The control system mainly consists of a PI speed controller, a PI current controller, a commutation controller, a pulse generator and so on.

As shown in Figure 8, the current sensor collects the current of phase A and phase B and calculates the non-commutated phase current *i*. The Hall sensor captures the moment when the commutation period starts through the collected position information of the brushless DC motor and combines the outgoing phase current to calculate the commutation signal required during the commutation period.

At the same time, the actual speed *n* of the motor can be calculated by the Hall sensor. The difference between the reference speed *n** and the actual speed *n* is the given value of the PI speed controller, the current *i** is the output of the PI speed controller, and the difference between the reference current *i** and the non-commutation current *i* is the input of the PI current controller. The duty cycle *d*(*d*_A_, *d*_B_, *d*_C_) is the output of the PI current controller. The output of the duty cycle should be selected according to the present voltage zone, and a double closed-loop of speed and current is formed to maintain the normal operation of the motor.

It can be seen from Figure 8 that the duty cycle inputting the pulse generator is selected according to the commutation signal. When the motor runs in the normal conduction period, the commutation signal is not generated, and the duty cycle *d*(*d*_A_, *d*_B_, *d*_C_) is input to the pulse generator. When the motor runs in the commutation period, the commutation signal is generated and the duty cycle *d*_com_ is input to the pulse generator. Then, the pulse generator generates the pulses of the DC-link switch and the pulses of the inverter switch to drive the motor system to work normally and to realize the reduction of commutation torque ripple of the motor.

## 4. Experimental Results and Analysis

In order to verify the correctness of the theoretical analysis and the effectiveness of the proposed strategy, an experimental platform was built. In the experimental platform, the AC power source is provided by a 6813C Keysight, the single-phase diode rectifier uses PB3510, the inverter and IGBT of DC-link switch use FGA25N120AN, and the motor load is provided by a Magtrol motor test system. Table 2 lists the BLDCM parameters and the AC power parameters of the experimental platform. According to the BLDCM parameters in Table 2, *U*_ab_ = 185 V in Equation (4) can be calculated.

Since the electromagnetic torque is proportional to the non-commutation current, the electromagnetic torque ripple can be approximately calculated to the non-commutation current ripple, and according to section 3.43 of IEC 60034-20-1, the torque ripple rate *K*_rT_ is defined as
(17)$$K_{\mathrm{rT}}=\frac{T_{\mathrm{high}}-T_{\mathrm{low}}}{T_{\mathrm{high}}+T_{\mathrm{low}}}\times 100\%$$
where *T*_high_ and *T*_low_ are the maximum and minimum torque in a period of time, respectively.

### 4.1. Experimental Results and Analysis at High Speed

When the motor runs at a high speed, the amplitude of the phase back-EMF is larger. The DC-link capacitor voltage may not satisfy Equation (16), which will result in failure to reduce the commutation torque ripple or an unsatisfactory reduction effect. Therefore, the boost control strategy proposed in this paper can improve the DC-link voltage to satisfy Equation (16).

Figure 9 shows the experimental results of the traditional control strategy and the proposed control strategy, respectively, under the high-speed condition (600 r/min, 0.89 N·m). In Figure 9, the experimental results from top to bottom are the three-phase current, the DC-link voltage *U*_D_link_, the DC-link capacitance voltage *U*_cap_, the electromagnetic torque and the commutation signal com.

Figure 9a shows the experimental waveform of the traditional control strategy. In the traditional control strategy, it can be seen from Figure 9a that the DC-link voltage waveform changes periodically and that the maximum voltage of the DC-link capacitor is only maintained at the amplitude of AC power source. Since the traditional strategy does not reduce the commutation torque ripple, the non-commutation current rippled significantly during the commutation period, which caused the motor torque ripple greatly. The motor torque ripple rate *K*_rT_ calculated through Equation (17) was 31.76%.

Figure 9b shows the experimental waveform of the proposed control strategy. In the proposed control strategy, the boost control mode is adopted in Zone B. It can be seen from Figure 9a that the DC-link capacitor voltage increased gradually and the non-commutation current ripple decreased significantly during the commutation period. The motor torque ripple rate *K*_rT_ calculated through Equation (17) was 25.53%.

From the comparison experiment in Figure 9a,b, under the same operation condition, the proposed strategy had no influence on the operation of the motor. Moreover, the proposed strategy made the three-phase current waveform more stable during the motor operation period. Through Equation (17), the motor torque ripple rate *K*_rT_ can be calculated. According to the calculation results, the torque ripple of the motor was significantly reduced under the proposed strategy.

In order to more clearly demonstrate the effectiveness of the proposed method in maintaining the stability of non-commutation current and reducing commutation torque ripple, the phase current, DC-link voltage and commutation signal waveforms of the traditional control strategy and the proposed control strategy in Figure ere arwe amplified. The blue block area was selected for zooming in Figure 9 and the amplified waveforms are shown in Figure 10.

Figure 10a,b are, respectively, the amplified waveforms of Figure 9a,b during the same commutation period. The experimental waveforms shown in Figure 10 from top to bottom are, respectively, the three-phase current of the motor, the DC-link voltage *U*_D_link_ and the commutation signal com.

As shown in Figure 10a, in the traditional control strategy there is a great difference between the rising rate of the motor’s positive conducting phase current *i*_A_ and the falling rate of the outgoing phase current *i*_C_ during the commutation period, thus resulting in a non-commutation phase current *i*_B_ with a great ripple. It can also be seen from Figure 10a that the commutation time of the motor is longer in the traditional control strategy.

As shown in Figure 10b, in the proposed control strategy, due to the DC-link voltage boost control mode, it can be seen the DC-link voltage waveform in Zone B is in a pulse pattern. During the commutation period, the rising rate of the positive conducting phase current *i*_A_ is basically consistent with the falling rate of the outgoing current *i*_C_, thus resulting in a non-commutation phase current *i*_B_ with less ripple. It can also be seen from Figure 10b that the commutation time of the motor is effectively reduced in the proposed control strategy.

From the comparison experiment in Figure 10a,b, compared with the traditional control strategy, the proposed strategy can effectively maintain the same rising rate of the positive conducting phase current and the falling rate of the outgoing current during the commutation period, thus maintaining the stability of the non-commutative current. At the same time, the commutation time of the motor was further shortened.

### 4.2. Experimental Results and Analysis at Low Speed

When the motor runs at low speed, the amplitude of the phase back-EMF is small. The DC-link capacitor voltage meets the high voltage required during commutation in the whole cycle, and the condition of Equation (16) is satisfied. At this time, the commutation torque ripple can be reduced without the DC-link boost control of the brushless DC motor.

Figure 11 shows the experimental results of the traditional strategy and the proposed strategy, respectively, when the motor runs at the low speed condition (200 r/min, 0.89 N·m). The experimental waveforms shown in Figure 11 from top to bottom are the three-phase current, the DC-link voltage *U*_D_link_, the DC chain capacitance voltage *U*_cap_, the electromagnetic torque Tor and the commutation signal com.

Figure 11a shows the experimental waveform of the traditional control strategy. In the traditional control strategy, it can be seen from Figure 9a that the DC-link voltage waveform changes periodically and the maximum voltage of the DC-link capacitor is only maintained at the amplitude of AC power source. During the commutation period, the non-commutation current has a significant ripple and the motor torque ripple rate *K*_rT_ calculated through Equation (17) is 24.3%.

Figure 11b shows the experimental waveform of the proposed control strategy. In the proposed control strategy, the DC-link voltage met the voltage of motor demand during the commutation period at low speed operation. Thus, the boost control mode was not adopted. It also can be seen from Figure 11b that the DC-link voltage waveform changed periodically and that the maximum voltage of DC-link capacitor was only maintained at the amplitude of the AC power source. The commutation torque ripple can be reduced simply by opening the DC-link switch during the commutation period. The motor torque ripple rate *K*_rT_ calculated through Equation (17) was 22.7%.

By comparing the above experimental results, it can be found that, in the low speed operation condition, the commutation frequency of the motor decreased with the decrease to the motor speed. Compared with the traditional control strategy, the non-commutation current fluctuation of the motor was improved and the torque fluctuation of the motor was reduced. At the low speed operation condition, the motor commutation frequency decreased. The proportion of torque ripple caused by the motor commutation was cut down and the torque ripple of the brushless DC motor was mainly caused by the ripple of the DC-link power source. Therefore, the proposed commutation torque ripple reduction strategy in this paper had no obvious effect at the low speed operation condition.

### 4.3. Controller Design

In order to verify the dynamic response performance of the strategy proposed in this paper, the experimental test platform was used to carry out the dynamic response experiment with the speed rising from 200 r/min to 600 r/min, and the load torque was always at 0.89 N·m.

Figure 12 shows the dynamic experimental results of motor speed under the proposed control strategy. The experimental waveforms from top to bottom are motor reference speed *n**, actual speed *n*, motor three-phase current, DC-link voltage *U*_D_link_ and DC-link capacitance voltage *U*_cap_.

Figure 12a shows the dynamic experimental results of the motor speed rising from 200 r/min to 600 r/min. It can be seen that the actual motor speed *n* changed with the change to the reference speed *n**, and the change rate was consistent with the reference speed *n**. The motor dynamic operation is very stable.

Figure 12b shows zoomed-in waveforms of dynamic experimental results of the motor speed in Figure 12a. As the motor speed increased, the amplitude of the phase back-EMF increased, and the DC-link capacitor voltage did not meet Equation (16). It can be seen from the Figure 12 that when the motor speed rises to about 400 r/min approximately, the DC-link boost energy storage mode begins in Zone B, and the DC-link capacitor meets the high voltage required during commutation period. The commutation torque ripple can be reduced.

## 5. Conclusions

Based on the BLDCM system with a DC-link small capacitor powered by an AC power source, this paper proposes a control method to reduce the commutation torque ripple using DC-link voltage boost technology and verifies the effectiveness of the proposed control method through experiments. The advantages of the proposed control method are as follows:(1)There is no need to set up an auxiliary boost circuit, and it is completed only by the characteristics of the motor inductance so that the DC-link capacitor can meet the high voltage required during the commutation period of the brushless DC motor.(2)The proposed strategy in this paper reduces the commutation torque ripple of the motor while maintaining the normal operation of the motor. Compared with traditional methods, the influence of the commutation torque ripple on the operation of the motor as reduced, and the stability of the motor system was improved. The proposed strategy is beneficial to the application of brushless DC motors in the field of high precision and high stability requirements.

## Figures and Tables

**Figure 1 micromachines-13-00226-f001:**
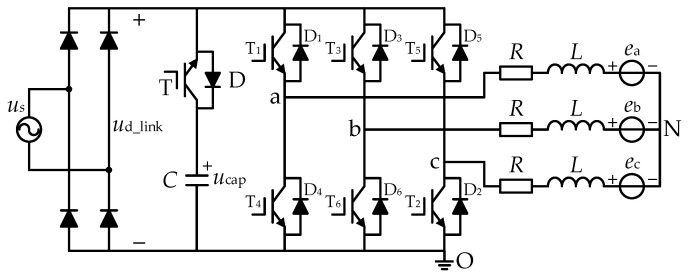
Topology of a brushless DC motor (BLDCM) drive system with a small DC-link capacitor.

**Figure 2 micromachines-13-00226-f002:**
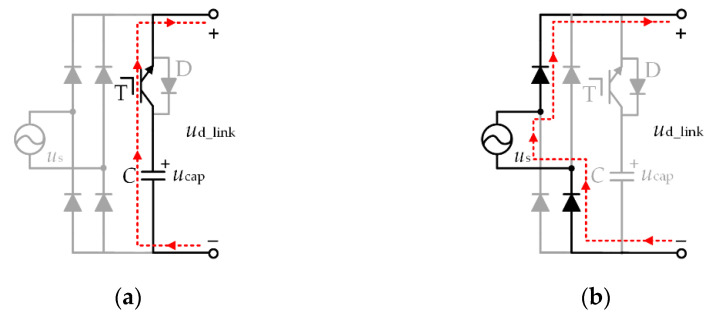
Equivalent circuit of the DC-link (**a**) switch T ON (**b**) switch T OFF.

**Figure 3 micromachines-13-00226-f003:**
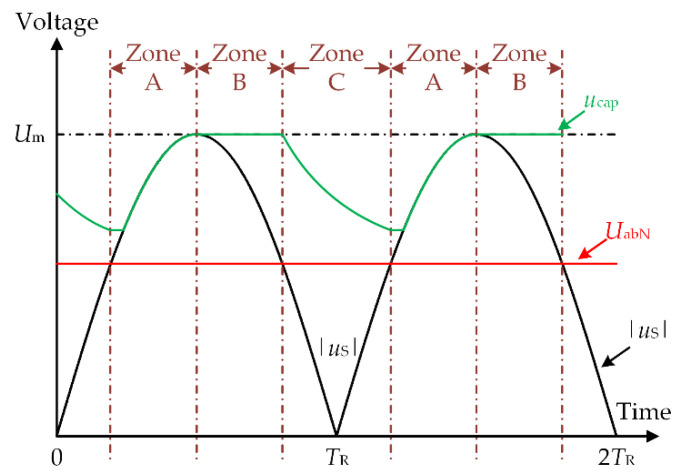
Waveforms of the capacitor voltage and the rectified mains supply voltage.

**Figure 4 micromachines-13-00226-f004:**
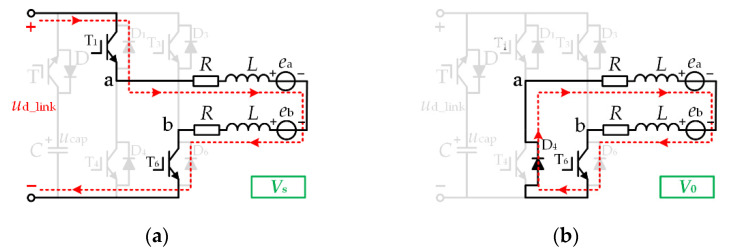
Vector equivalent circuit (**a**) *V*_s_ (**b**) *V*_0_.

**Figure 5 micromachines-13-00226-f005:**
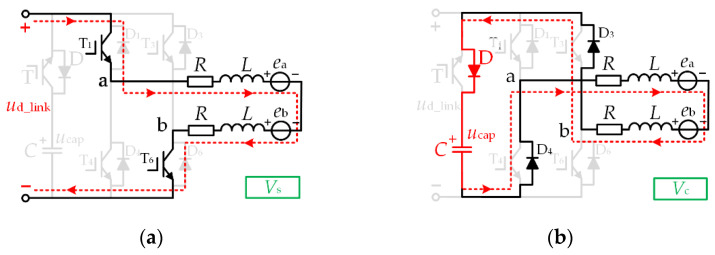
Vector equivalent circuit (**a**) *V*_s_ (**b**) *V*_c_.

**Figure 6 micromachines-13-00226-f006:**
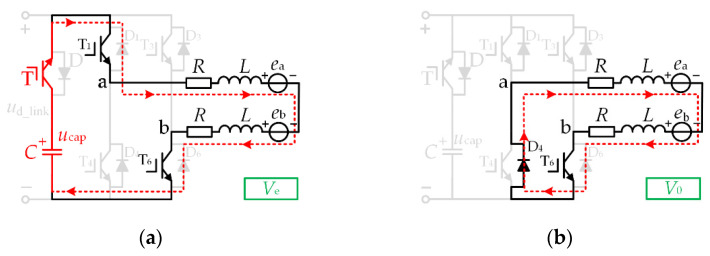
Vector equivalent circuit (**a**) *V*_e_ (**b**) *V*_0_.

**Figure 7 micromachines-13-00226-f007:**
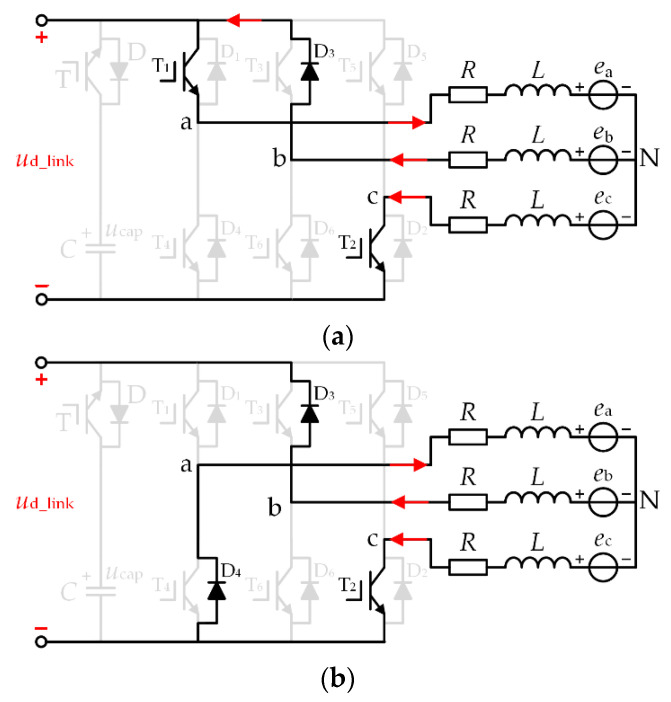
Equivalent circuit during the “a + b − → a + c −” commutation period (**a**) switch T_1_ ON (**b**) switch T_1_ OFF.

**Figure 8 micromachines-13-00226-f008:**
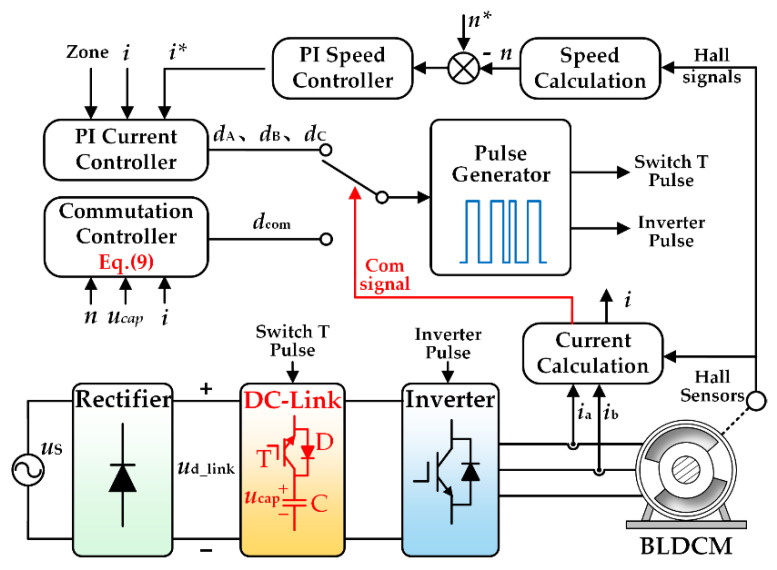
A system control structure diagram of the strategy is proposed.

**Figure 9 micromachines-13-00226-f009:**
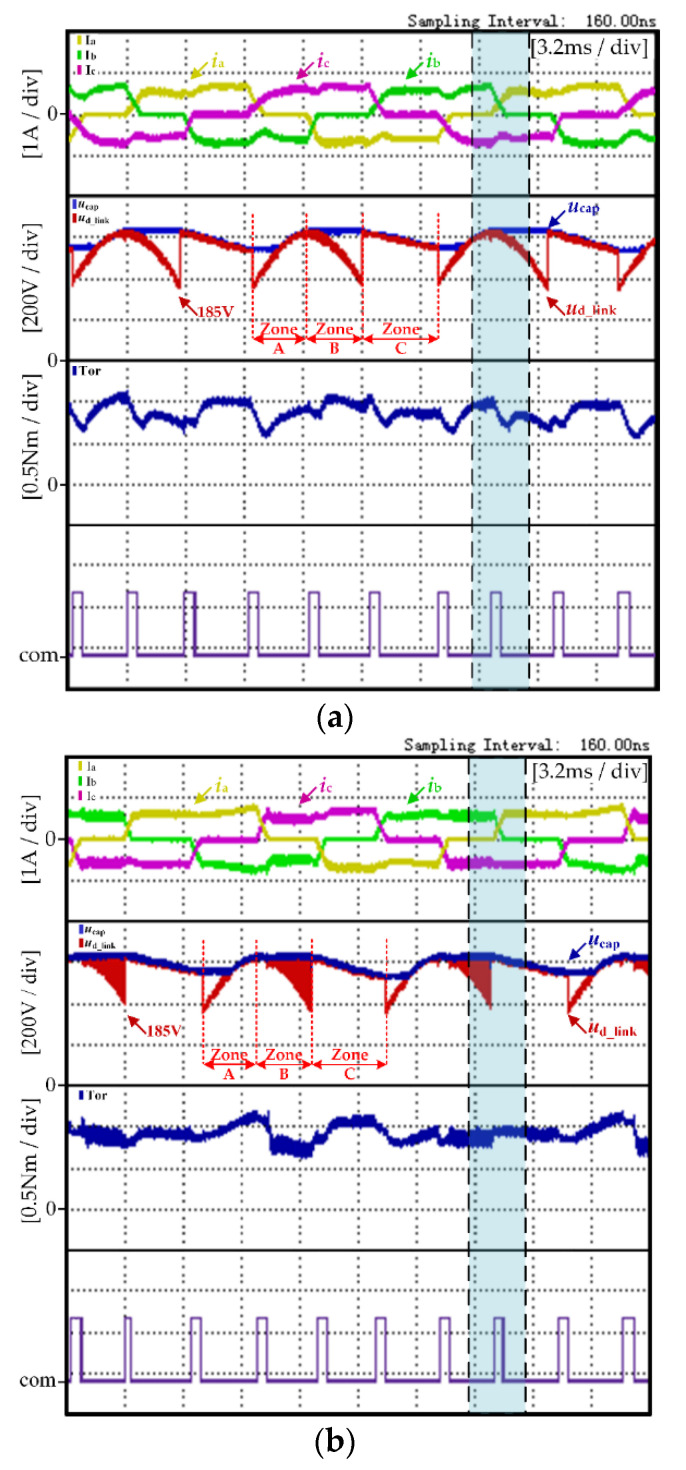
Experimental results under the high-speed condition (600 r/min, 0.89 N·m): (**a**) traditional strategy (**b**) proposed strategy.

**Figure 10 micromachines-13-00226-f010:**
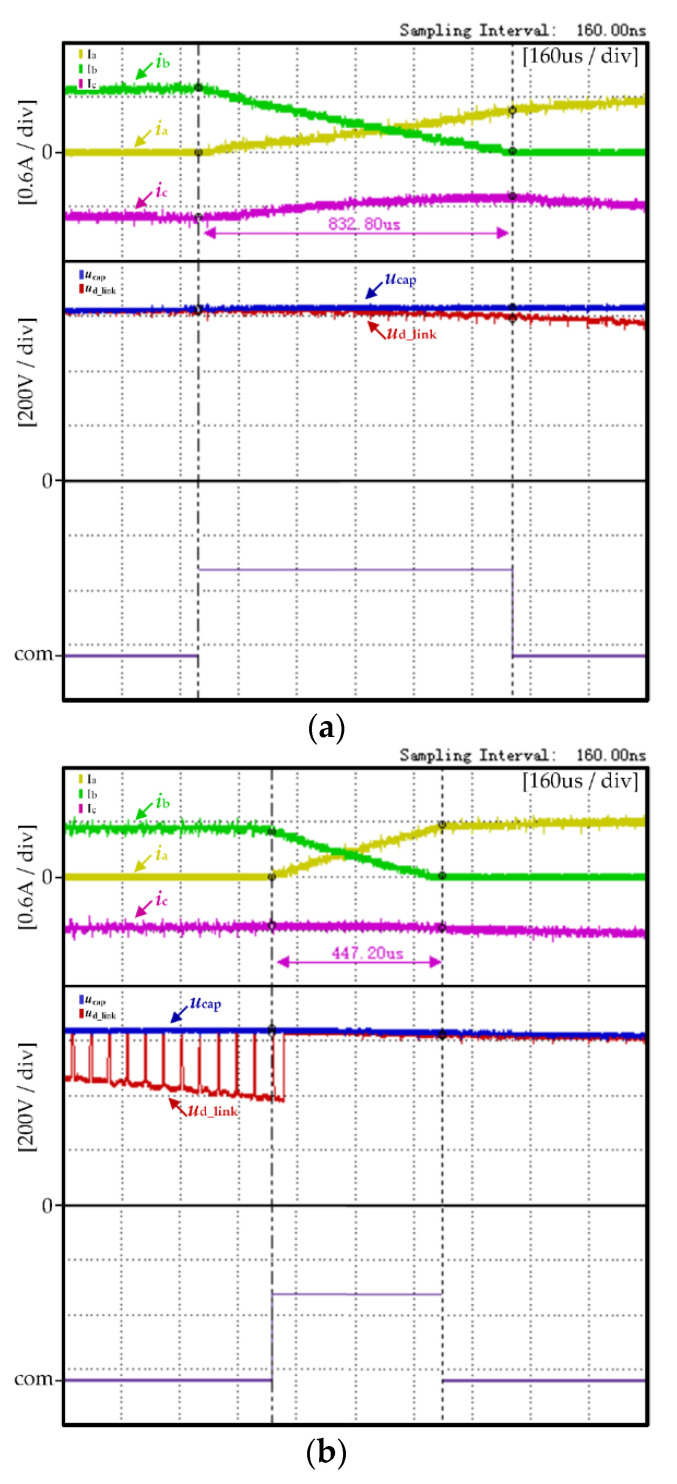
Enlarged view of the experimental results under the high-speed condition (600 r/min, 0.89 N·m): (**a**) traditional strategy (**b**) proposed strategy.

**Figure 11 micromachines-13-00226-f011:**
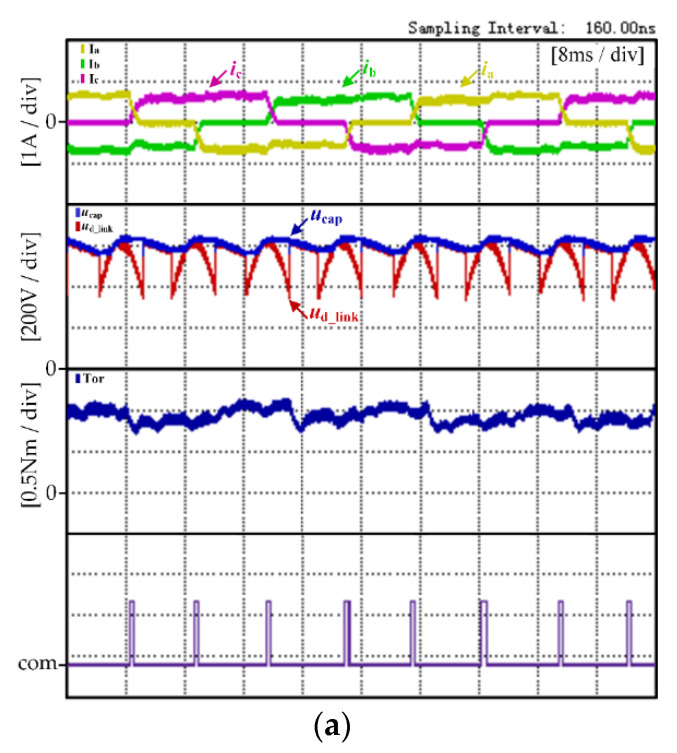
Experimental results under the low speed condition (200 r/min, 0.89 N·m): (**a**) traditional strategy (**b**) proposed strategy.

**Figure 12 micromachines-13-00226-f012:**
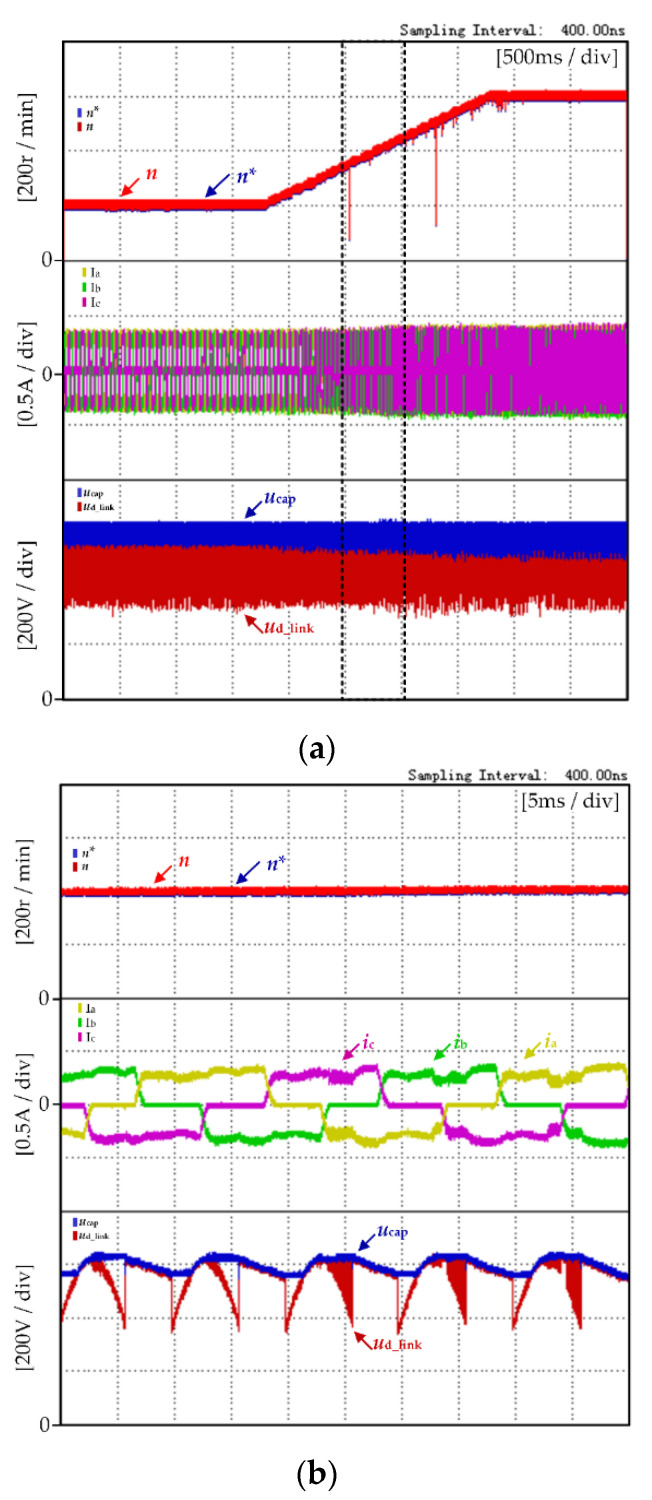
Dynamic response of speed variation in the proposed strategy (*T*_L_ = 0.89 N·m): (**a**) overall view; (**b**) enlarged view.

**Table 1 micromachines-13-00226-t001:** The state of DC-link switch T and inverter switch T_1_, T_6_ during the “a + b −” normal conduction period.

Zone	State of T	State of T_1_	State of T_6_
A	OFF	*d* _A_	ON
B	OFF	*d* _B_	ON
C	ON	*d* _C_	ON

**Table 2 micromachines-13-00226-t002:** The BLDCM system parameters and AC power parameters of the experimental platform.

	Parameter	Symbol	Value
BLDCM	Rated current	*I* _N_	0.6 A
	Rated back EMF	*E* _N_	65.5 V
	Phase resistance	*R*	44.9 Ω
	Phase inductance	*L*	120 mH
	Back EMF coefficient	*K* _e_	0.7825 V/(rad/s)
	Rated load	*T* _N_	0.89 N·m
	Rated speed	*n* _N_	800 r/min
	Poles pairs	*P*	5
Main Supply	Mains supply voltage	*U* _s_	220 Vrms
	Mains supply frequency	*f*	50 Hz

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
