# Peer review of "Commutation Torque Ripple Reduction Strategy of Brushless DC Motor Drives Based on Boosting Voltage of DC-Link Small Capacitor"

_micromachines, 2022, doi:10.3390/mi13020226_

Round 1

Reviewer 1 Report

The author(s) proposed the manuscript on “Commutation Torque Ripple Reduction Strategy of Brushless DC Motor Drives Based on Boosting Voltage of DC-link Small Capacitor”. It is very difficult to follow the manuscript. However, the following queries should be addressed before further consideration of the manuscript:

  1. Rewrite the abstract to highlight the findings of contribution.
  2. In the introduction, the key objectives of the manuscript should be provided in order to improve the readability of the manuscript.
  3. The manuscript contains some typographical and grammatical mistakes which should be corrected.
  4. The figures should be presented in better resolution.
  5. As written in section 4.1, “Figure. 10a and 10b are respectively the amplified waveforms of Figure 9a and 9b during the commutation period”. For this figure, amplified waveforms are not properly explained and presented.
  6. The result analysis should be discussed in more detail along with comparative analysis.
  7. The author(s) should justify what is novelty in this manuscript.
  8. It is sometimes very difficult to follow the contents of a manuscript. Please improve the continuity, readability and overall language, and flow of technical content.

Author Response

Dear editors and reviewers:

Thanks for your letter and the constructive comments of the reviewers concerning our manuscript entitled “Commutation Torque Ripple Reduction Strategy of Brushless DC Motor Drives Based on Boosting Voltage of DC-link Small Capacitor” (micromachines - 1552024). Those comments are valuable and very helpful for revising and improving our paper. We have thoroughly considered all the comments and made careful modifications to the original manuscript. We will respond to each comment point by point, along with a clear indication of the location of the revision. We will highlight the changes with yellow color in the revised manuscript. Please see the attach file.

Thank you!

Reviewer 2 Report

This paper proposed a DC-link boost control strategy to reduce motor commutation torque ripple based on the brushless DC motor system with small DC-link capacitor. It is an interesting topic. This paper can be improved by considering the comments as below.

  1. The actual Figure 3 at line 93 is missed.
  2. What are the definitions of effective vector, charging vector, and zero vector?
  3. There is no description in the text on Table 1 and 2.
  4. Please make sure the correctness of State of T in Table 1.
  5. The equation numbers are not correct at lines 207, 211, and 212.
  6. Please check the symbol of duty cycle at lines 201 and 21.
  7. It is better to show the results at the same commutation period for comparison in Figure 10.
  8. The case of variable subscripts should be consistent.
  9. The torque ripple rate is reduced only from 3% to 22.7% at speed of 200 rpm by the proposed algorithm. Is it necessary to improve it by including some method or modifying the proposal to expand the operation range?

Author Response

(The authors gave the same response as above.)
